# Simulation and Optimization of Optical Fiber Irradiation with X-rays at Different Energies

**Arnaud Meyer** [1] ID, **Damien Lambert** [2] ID, **Adriana Morana** [1] ID, **Philippe Paillet** [2], **Aziz Boukenter** [1] ID and **Sylvain Girard** [1,*]

1   Laboratoire Hubert Curien, UMR-CNRS 5516, Université Jean Monnet, F-42000 Saint-Etienne, France; arnaud.meyer@univ-st-etienne.fr (A.M.); adriana.morana@univ-st-etienne.fr (A.M.); aziz.boukenter@univ-st-etienne.fr (A.B.)
2   CEA, DAM, DIF, 91297 Arpajon, France; damien.lambert@cea.fr (D.L.); philippe.paillet@cea.fr (P.P.)
*   Correspondence: sylvain.girard@univ-st-etienne.fr

**Simple Summary:** We investigated the influence of modifying the voltage of an X-ray tube, and therefore its photon energy spectrum, on the Total Ionizing Dose deposited in a single-mode, radiation sensitive, optical fiber. Simulation data, obtained using a toolchain combining SpekPy and Geant4 software, are compared to experimental results and demonstrate an increase of the deposited dose with operating voltage, which is mainly caused by low-energy photons below 30 keV.

**Abstract:** We investigated the influence of modifying the voltage of an X-ray tube with a tungsten anode between 30 kV and 225 kV, and therefore its photon energy spectrum (up to 225 keV), on the Total Ionizing Dose deposited in a single-mode, phosphorus-doped optical fiber, already identified as a promising dosimeter. Simulation data, obtained using a toolchain combining SpekPy and Geant4 software, are compared to experimental results obtained on this radiosensitive optical fiber and demonstrate an increase of the deposited dose with operating voltage, at a factor of 4.5 between 30 kV and 225 kV, while keeping the same operating current of 20 mA. Analysis of simulation results shows that dose deposition in such optical fibers is mainly caused by the low-energy part of the spectrum, with 90% of the deposited energy originating from photons with an energy below 30 keV. Comparison between simulation and various experimental measurements indicates that phosphosilicate fibers are adapted for performing X-ray dosimetry at different voltages.

**Keywords:** optical fibers; X-ray tubes; Geant4; radiation effects; dosimetry

## 1. Introduction

### 1.1. Interest of X-rays for Radiation Testing

Radiation testing can involve a variety of ionizing radiation sources, such as photons, protons, electrons, neutrons, or heavy ions. The choice of a certain type of radiation source depends on multiple factors, including conformity to a target environment, emphasis on certain physical processes, and observation of standard practices.

Availability and ease of use are other factors that play a role in the actual planning of such radiation testing. In this regard, X-ray tests have significant advantages over other kinds of radiation sources. X-ray tubes, in particular, have been used for over a century for various applications, ranging from medical imaging [1] to material characterization [2]. These sources of high-energy photons, typically up to several hundreds of keV, are available commercially and therefore relatively easy to procure, install and manipulate safely, compared, for instance, to radioisotope sources.

A typical X-ray tube contains a cathode and an anode, both sealed in a vacuum. The cathode is typically a filament through which a very small electrical current circulates, on the order of several mA. A very high voltage, on the order of tens to hundreds of kV, is

applied between the cathode and the anode, causing electrons extracted from the cathode to be accelerated at very high velocity towards the anode, effectively forming an electron beam. Finally, the anode, typically a thick layer of a high-Z material like tungsten, causes the conversion of part of the incoming electron beam to photons through two physical processes: bremsstrahlung, generating a continuous energy spectrum until a threshold determined by the tube voltage; and characteristic emission, generating very intense and narrow energy peaks characteristic of the anode material. The beam exiting the X-ray tube is therefore a combination of these two processes: a continuous energy spectrum along with sharp characteristic peaks [1–3]. This beam is emitted in every direction in space, but practical limitations, such as the orientation of the anode and the presence of an output window on the X-ray tube, cause it to be limited to a cone of radiation originating from the anode.

Between the X-ray source itself and the sample being irradiated, several interceding elements cause a modification of both the energy spectrum and intensity of the X-ray beam. First, as the anode can be assimilated to a point source, the intensity of the beam decreases naturally with increasing distance from the tube, following a reverse square law. Second, there are numerous materials between the point of emission and the sample, including the window of the X-ray tube, typically made of a low-Z material such as beryllium, and a layer of air, both significantly absorbing very-low-energy photons. Additional filtration, typically from materials like aluminum, can also be considered to reduce even more the low-energy part of the spectrum, causing the mean energy of the beam to increase, which can optimize dose deposition in thick samples [4].

## 1.2. Importance of Dosimetry and Its Accuracy

Absorbed dose is a key quantity in applications that involve the presence of ionizing radiation, and dosimetry is the measurement of this quantity. The International Commission on Radiation Units and Measurements (ICRU) defines absorbed dose as the quotient between the mean energy imparted by ionizing radiation to a sample of matter and the mass of this sample. The unit of absorbed dose is J/kg, which is also given the special name Gray (Gy) [5]. Because radiation interacts in different ways and intensities with different materials, it is common in dosimetry to specify the material for which a quantity of absorbed dose is applicable by including the name of the material in the unit of the result, appearing as Gy(material).

In the domain of radiation damage applied to materials, the absorbed dose is usually categorized in two different families of processes: the Total Ionizing Dose (TID) relates to the dose due to ionization events [4,6], whereas the Displacement Damage Dose (DDD) refers to the dose due to the displacement of atoms, and is of particular significance in crystalline materials like semiconductors [7,8].

Improvement of the accuracy of dosimetry is an important topic in all applications where such measurements are needed, despite the variety of radiation environments and types of dosimetry devices involved. Research towards more accurate dosimetry crosses many scientific fields and applications, including radiotherapy [9,10], radiation protection [11], radiation testing of electronic devices [4,6–8], space missions [12,13], and even large physics instruments [14–16].

Because the physical framework of dosimetry involves particle interaction at the atomic level, the need to improve the understanding and accuracy of the dose deposition process and its measurement brought forward simulation tools to reproduce as accurately as possible these physical processes. Because of their ability to simulate individual particles and events, Monte-Carlo codes have become one of the tools of reference due to their efficiency and consistency to perform dosimetry calculations [17–19].

## 1.3. Use of Optical Fibers as Dosimeters

Silica-based optical fibers (OFs) are passive waveguides that operate at optical wavelengths, typically between the ultraviolet (~300 nm) and infrared (~2000 nm) domains [20].

Their functioning principle relies on a difference of refractive index between the central element of the fiber, named *core*, and its surrounding element, named *cladding*. In practice, both these elements are covered by a protective, polymer-type material, named *coating*, that does not play a role in its guiding properties. The typical base material of such OFs is amorphous silica $SiO_2$, doped differently between core and cladding in order to achieve the desired refractive index contrast [21]. In terms of radiation behavior, these dopants play an important role, and can make the OF range anywhere between radiation-hardened to radiation-sensitive [22]. Radiation effects on optical fiber are usually categorized in three areas: Radiation-Induced Attenuation (RIA) causing the transmitted signal to decrease under radiation; Radiation-Induced Emission (RIE) causing light to be emitted inside the OF under radiation; and Radiation-Induced Refractive Index Change (RIRIC) causing the refractive index of the fiber material to be modified under radiation [23].

OFs have emerged as a promising technology for dosimetry because of their relative immunity to external electromagnetic radiation, their ability to be used both as a sensitive element and a means to transport signal, as well as for their low dimensions which enable space-resolved measurements or access to space-constrained applications. The use of OF probes for dosimetry in radiation therapy is an increasing domain of research, using different interrogation techniques [24,25] as well as different types of OFs.

Phosphorus (P)-doped OFs in particular exhibit strong radiation sensitivity and have been the object of ample research to assess their dosimetric properties [15,16,26–28]. More specifically, the RIA of P-doped OFs in the 1550 nm wavelength region—caused by the absorption band of $P_1$ defects induced in phosphosilicate glasses by high-energy radiation [20,29]—was shown to be mostly linear (within 5%) up to doses of 500 Gy [15], lowly sensitive (within 15%) to temperature between $-120$ and $80$ °C [27], and stable in time after irradiation [15]. Therefore, numerous studies have considered using P-doped OFs as dosimeters, especially combined with interrogation techniques allowing to map attenuation of an OF through its length, effectively resulting in distributed dosimeters [14,15]. Moreover, various radiation tests have shown that this RIA response scales very well over a wide variety of radiation sources, including steady-state X-rays [15,16,26–28,30,31], pulsed X-rays [30], $\gamma$ rays [32–34], protons [15], neutrons [31,32], and mixed field such as the CHARM facility in CERN [35].

These previous experimental results overall indicate that P-doped OFs are only sensitive to TID effects and relatively unaffected by DDD effects [31]. Although such a statement could be challenged in very high neutron fluence environments such as in-core instrumentation, for most cases, these previous research works hint towards the possibility of investigating and qualifying the radiation properties of OFs interchangeably between different radiation environments. In regard to the advantages described above, the use of X-ray generators for such preliminary research offers therefore a strong advantage in terms of accessible dose and dose rate ranges, budget, flexibility, reliability and safety.

### 1.4. Influence of X-ray Voltage on Optical Fiber Dosimetry

Despite the long-standing practice of X-ray irradiation, several practical questions remain regarding the use of such irradiators for qualifying the radiation response of OFs. Dosimetry, in particular, is a key element that depends on multiple factors, such as distance from the source, voltage and current of the X-ray tube, and material or geometry of the irradiated sample. The use of dosimetry devices, like ionization chambers, helps in providing an in situ measurement of the dose or dose rate at a certain functioning point. However, such dosimeters do not deliver a measurement directly corresponding to the irradiated sample, but rather use a standardized unit, such as dose in water or air kerma. In order to properly understand and optimize the irradiation process using X-rays, the correspondence between ionization chamber dosimetry and the actual dosimetry of the irradiated sample needs to be properly understood and modeled, especially regarding the wide range of energy spectra enabled by setting different X-ray tube voltages. The particular geometry of OFs—being long and extremely thin compared to ionization chambers that are

only able to provide measurements averaged through their sensitive volume—is another key difference that requires proper evaluation.

This study aims to explore and model the dosimetry of OFs irradiated using X-ray tubes in order to properly understand and optimize the irradiation process of such elements, as well as the influence of key parameters such as X-ray tube voltage, on the actual dosimetry of the samples.

## 2. Materials and Methods

### 2.1. Optical Fiber Irradiation

The actual OF irradiation experiments were carried out in the LabHX facility of Laboratoire Hubert Curien of Université Jean Monnet Saint-Étienne, France. This irradiator is equipped with a COMET MXR-225/26 X-ray tube operating up to 225 kV and a current up to 30 mA (20 mA at the highest voltage). This X-ray tube includes a tungsten anode with an angle θ of 30°, which center is located 4.3 cm above a 2 mm-thick sealing window made of beryllium, resulting in a nominal irradiation cone with an angle α of 40°.

A simplified schematic of the X-ray tube is shown in Figure 1, along with the definition of the coordinate system used in this paper, which is the same as the software SpekPy (cf. Section 2.2): x is orientated in the anode-cathode direction and positive towards the cathode, and z is the central axis and positive in the X-ray beam propagation direction. The direction and orientation of the y axis can be determined using the right-hand rule.

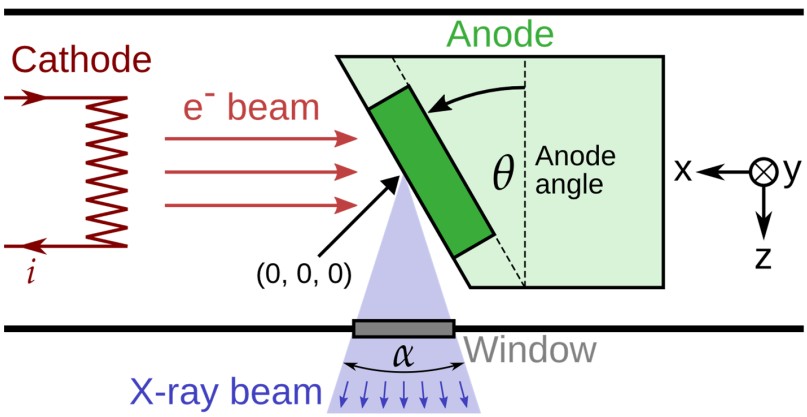

**Figure 1.** Simplified schematic of a typical X-ray tube and of the coordinate system of this study.

The irradiation setup is summarized in Figure 2. Each irradiated sample was taken from a P-doped OF manufactured by iXblue. Every OF sample was cut to a length of 1 m and coiled in a flat spiral of 5 cm internal radius in order to reduce as much as possible its size, and therefore beam deviation, while keeping a high enough bending radius to ensure good guiding of the signal inside the OF.

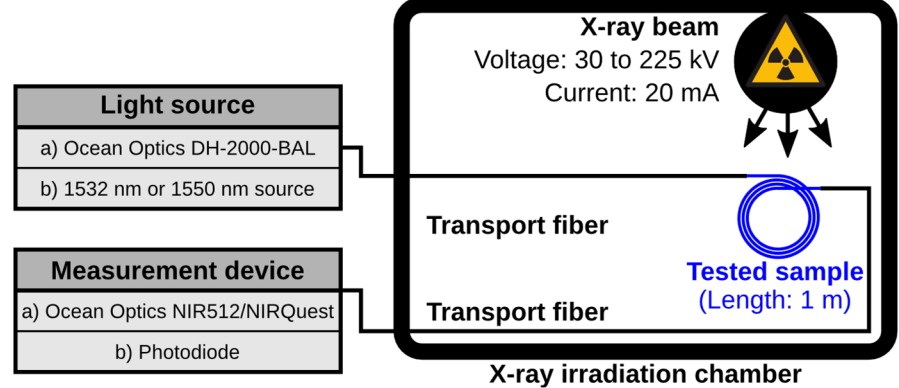

**Figure 2.** Experimental setup of OF irradiation with an X-ray tube at different voltages.

Each OF sample was spliced at both ends to a transport, radiation-hardened OF, connecting it to instrumentation placed outside of the irradiation chamber. One end was connected to a light source, the other end to a measurement device.

Two kinds of measurements were performed in this setup: spectral measurements were acquired using an Ocean Optics DH-2000-BAL deuterium-halogen light source, covering a continuous spectrum between wavelengths of 210 nm and 2500 nm; and two models of compact infrared spectrometers: Ocean Optics NIR512, operating between 856 nm and 1735 nm, and Ocean Optics NIRQuest, operating between 900 nm and 2137 nm. Further RIA measurements were performed using an optical source at either 1532 nm or 1550 nm and a photodiode to measure the transmitted optical power.

The samples were centered around the following coordinates: (y = 0, z = 42.5 cm), and were positioned on the x-axis to be centered around the beam maximum as described in Section 3.1. They were irradiated at room temperature (between 18 °C and 28 °C) at five different X-ray tube voltages between 30 kV and 225 kV, and a constant X-ray tube current of 20 mA, for a time period between 900 s and 3600 s for each OF sample.

Dosimetry was performed using a PTW 23344 ionization chamber connected to a UNIDOS E reading unit, which is calibrated for dose in water and therefore delivers a dose rate reading in Gy($H_2O$)/s. The plane-parallel ionization chamber we used has a sensitive volume diameter of 15.9 mm, which only enables a coarse spatial resolution. Its documentation also states that this ionization chamber is optimized for use with X-ray tube voltages from 15 kV to 70 kV, although in this study we evaluated the raw, uncorrected measurement of the dosimetry system from 30 kV to 225 kV.

## 2.2. Simulation of X-ray Spectrum and Fluence Rate

In order to determine the X-ray spectrum and fluence rate as it reaches the irradiated OF, we programmed a simulation based on the SpekPy software.

SpekPy [36] is a Python library that models the spectrum of X-ray tubes. It is the successor of the stand-alone software SpekCalc [37]. SpekPy is able to calculate the spectrum, but also key parameters like fluence, half-value layer or air kerma, at any position from a defined X-ray source. It handles tungsten (W) anodes operated at voltages between 30 and 300 kV, as well as molybdenum (Mo) and rhodium (Rh) between 20 and 50 kV. It also features the functionality to accurately simulate the filtration of materials standing between the source and the sample, which takes into account the increase of filtration path observed for off-axis measurements. The accuracy of SpekPy was verified against standard NIST X-ray spectra [38].

In this work, we used SpekPy v2.0.8 (last updated in May 2022) with the *kqp* physics model, which is described by SpekPy authors to be the most accurate, especially regarding off-axis estimations [36]. The use of this physics model had notably a strong influence on the results of Section 3.1.

## 2.3. Simulation of Deposited Dose in Optical Fiber

In order to obtain the dose deposited by X-rays in the different parts of the irradiated OF, we programmed a simulation based on the Geant4 software.

Geant4 [39–41] is a C++ toolkit that provides all the necessary elements to build a Monte-Carlo simulation of particle physics over a wide domain of energies, scales and applications. It was first developed by CERN and is now maintained by an international collaboration [39], which provides regular updates to its core and physics engines.

In this work, we used Geant4 v11.1 (released in December 2022), along with the *QBBC_EMZ* physics package, which is the recommended package for the simulation of radiation effects in a space environment [42]. It includes the *G4EmStandardPhysics_option4* physics module, which uses the most accurate electromagnetic models and tracking of charged particles, especially at low energies [43].

The P-doped OF was modeled by three cylinders (*G4Tubs* objects) for the core, cladding and coating, respectively, using the diameters mentioned in Figure 3 and a common length

of 1 mm. Each layer was modeled by a different material: $SiO_2$ with 6.6 wt% concentration of phosphorus and density 2.21 g/cm$^3$ for the core, pure $SiO_2$ with density 2.20 g/cm$^3$ for the cladding, and acrylate $C_5H_3N_1$ with density 1.18 g/cm$^3$ for the coating.

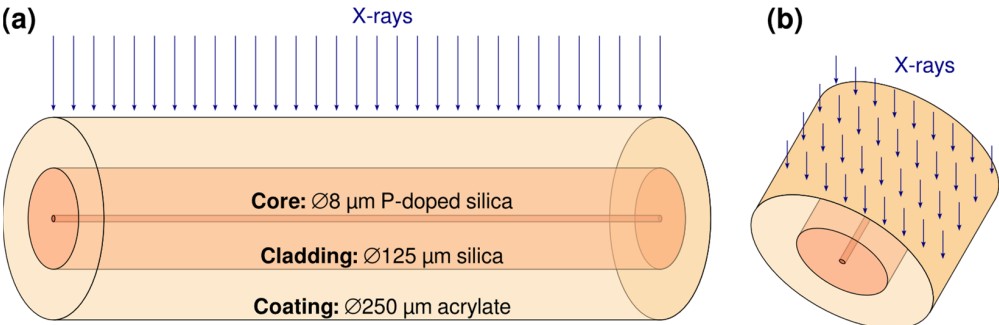

**Figure 3.** Geometry of the Monte-Carlo simulation of dose deposition in the irradiated OF, viewed (**a**) from the side, (**b**) from the top.

The incoming particles were configured using the *G4GeneralParticleSource* module, with a particle type set to photons, a rectangular source of dimensions 250 μm × 1 mm (the projected surface of the OF on a horizontal plane) placed 1 mm above the simulated OF, and a direction set vertically towards the OF. The spectrum of the particles was either monoenergetic from 1 keV to 2000 keV, or using the spectrum simulated by SpekPy, as described further in this article.

Dose deposition was determined using the *G4PSDoseDeposition* primitive scorer, which divides the energy deposited in a volume, determined by the physical processes implemented in the physics libraries, by the mass of this volume. Energy deposition is simulated discretely at every step of a *cut in range* (set to 1 nm in our case), and continuously in-between [39,41].

Each simulation was run for 1,000,000 photons, corresponding to a photon fluence of $4 \times 10^8$ cm$^{-2}$ per run, and the dose deposited in the core, cladding and coating of the OF were totalized for each run. Each of these runs was repeated 100 times with different random number generator seeds in order to evaluate the statistical deviation, and therefore the uncertainty, of the results.

### 3. Results and Discussion

#### 3.1. Shape of the X-ray Beam and Position of the Maximum

A first important parameter to consider with X-ray tubes is an offset of the beam, typically observed towards the anode to cathode direction. This offset, called the *heel effect* in the medical domain [44], is caused by the fact that physical phenomena causing the conversion of electrons to photons take place inside the anode material, and therefore newly produced photons need to go through different thicknesses of this material depending on their direction, resulting in an angular spectrum with a privileged direction. This direction of maximum beam intensity is typically off-axis, and can be estimated by simulation.

Figure 4a depicts the normalized X-ray fluence simulated by SpekPy according to the position on the x-axis, for 5 different operating voltages between 30 kV and 225 kV. It highlights a maximum that is always located off-axis, i.e., at an x position different than 0. As the X-ray tube voltage increases, the beam maximum moves even further from the axis.

Figure 4b shows a two-dimensional map of the simulated fluence rate for a single voltage of 100 kV. This image shows that although the beam is shifted several centimeters in the x-axis from the vertical of the X-ray anode (x = 0), it is symmetrical in the y-axis because the X-ray tube geometry is entirely symmetric in this axis. The composition of the heel effect in the x-axis and symmetry in the y-axis brings an elliptical shape to the actual beam spot. This figure also features the limit of the 40° emission cone of our setup, beyond which the fluence should be negligible in practice. For later reference, an outline of the dimensions of an irradiated OF is shown in the solid line, along with the location of the

four dosimetry points that will be considered in the rest of the document, each located 5 cm from the OF center.

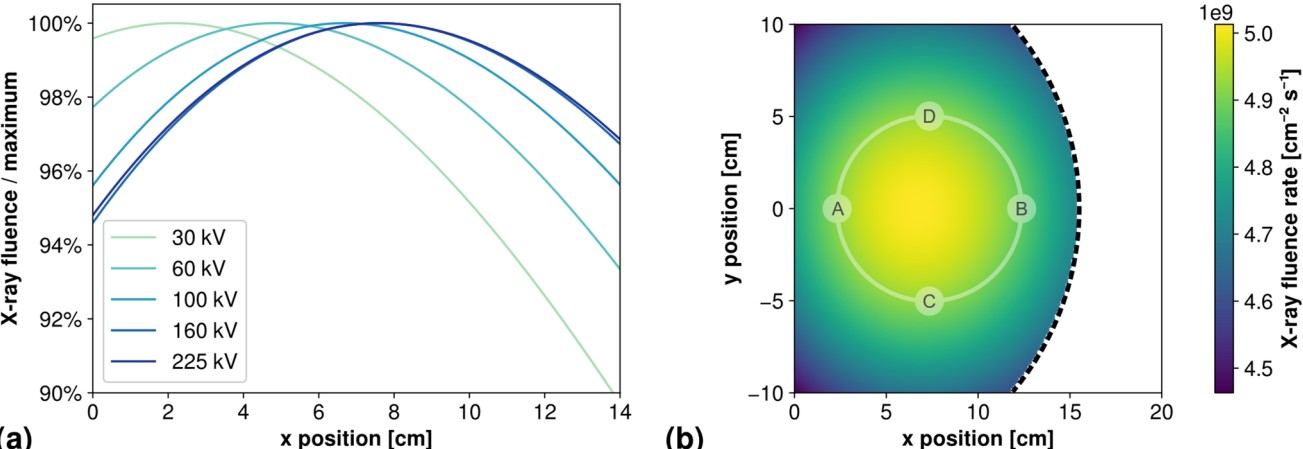

**Figure 4.** (**a**) Simulated fluence (normalized by maximum) of the X-ray beam according to position on the x-axis and X-ray tube voltage, at vertical position z = 42.5 cm. (**b**) Simulated fluence rate map of the X-ray beam at a position of z = 42.5 cm, tube voltage of 100 kV and current of 20 mA. Dashed contour shows the limits of the X-ray beam 40° cone, and solid contour shows the outline of an OF sample, with the location of the four dosimetry points A, B, C and D, each drawn inside a circle indicating the size of the ionization chamber sensitive surface.

Table 1 shows the simulated position of this beam maximum for the five investigated operating voltages, along with the experimental positions at which the OFs were placed, determined by ionization chamber measurements (with an uncertainty of $\pm 1$ cm). Both these measurements are in quite good agreement, except for the highest voltages at which the experimental position was set a few centimeters further than the simulated one. This could be due to an incorrect experimental estimation of the beam maximum position, also evidenced in the off-centering of ionization chamber measurements exposed in Section 3.6.

**Table 1.** Simulated x position of the X-ray beam maximum, and location of the irradiated OF sample, according to operating voltage.

| X-ray Tube Voltage | 30 kV | 60 kV | 100 kV | 160 kV | 225 kV |
|---|---|---|---|---|---|
| Simulated x position of beam maximum [cm] | 2.2 | 4.9 | 6.7 | 7.6 | 7.6 |
| Experimental x position of OF coil center [cm] | 2 | 5 | 7 | 9 | 10 |

### 3.2. X-ray Energy Spectrum, Mean Energy and Fluence at Different Tube Voltages

Using SpekPy we simulated the X-ray beam spectrum at the location of the optical fiber sample. Because of the circular shape of the sample, we considered four locations placed at 5 cm, as introduced in Section 3.1. The spectrum was simulated at these four locations, and we calculated the average of these four spectra to obtain the average spectrum and fluence observed by the irradiated OF. Uncertainty was calculated from the standard deviation between these four sets of data.

After inputting parameters corresponding to the COMET MXR-225/26 source, X-ray spectra were simulated in these conditions with tube voltage varying between 30 kV and 225 kV with a step of 5 kV, and a constant tube current of 20 mA. Filtration from the beryllium window of the X-ray tube and the layer of air between the window and the sample was considered in the simulation.

An excerpt of the resulting spectra is shown in Figure 5a, for five voltages between 30 kV and 225 kV. Both features of the X-ray spectrum are easily identifiable: the continuous background caused by the bremsstrahlung effect, with a cut-off energy corresponding to the operating voltage, and the sharp characteristic peaks of the tungsten anode. These main

characteristic peaks are identified on the figure by their X-ray emission spectroscopy line names, summarized in Table 2. Through all investigated voltages, characteristic emission accounts for between 27% and 38% of the total fluence.

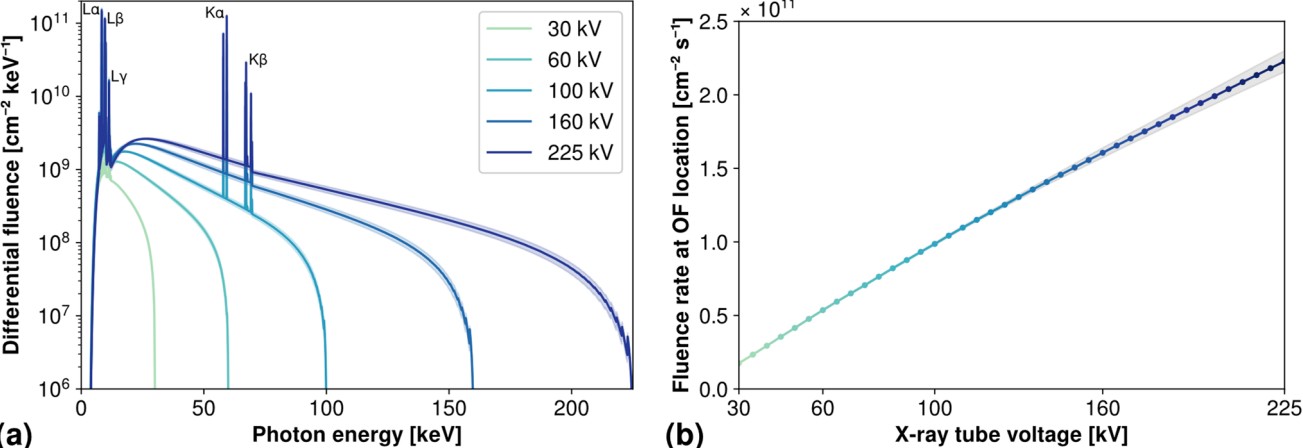

**Figure 5.** (**a**) Simulated energy spectra of the X-ray beam for five different source voltages from 30 kV to 225 kV at a location corresponding to the irradiated OF, with a constant current of 20 mA. The names of the main characteristic X-ray emission lines of tungsten are given next to the corresponding peaks. (**b**) Simulated fluence rate of the X-ray beam for 40 different voltages from 30 kV to 225 kV at locations corresponding to the irradiated OFs, with a constant current of 20 mA. Filled areas indicate uncertainties at $\pm 2\sigma$.

**Table 2.** Main characteristic X-ray emission lines for element tungsten (W) [45].

| X-ray Line Name | K$\alpha$ | K$\beta$ | L$\alpha$ | L$\beta$ | L$\gamma$ |
|---|---|---|---|---|---|
| Main electronic transitions | KL$_3$ (K$\alpha_1$) KL$_2$ (K$\alpha_2$) | KM$_3$ (K$\beta_1$) KN$_{2/3}$ (K$\beta_2$) | L$_3$M$_5$ (L$\alpha_1$) L$_3$M$_4$ (L$\alpha_2$) | L$_2$M$_4$ (L$\beta_1$) L$_3$N$_5$ (L$\beta_2$) | L$_2$N$_4$ (L$\gamma_1$) |
| Mean energy [keV] | 58.8 | 67.7 | 8.4 | 9.7 | 11.3 |

As the operating voltage increases, the spectrum shifts towards higher energies because of bremsstrahlung; but it should also be noted that the overall intensity increases with increasing voltage, even at lower energies, despite the operating current remaining at the same value of 20 mA.

Calculating the integral of such an energy spectrum gives the total fluence $\phi$ simulated at a considered location for a given operating voltage. This operation can be conveniently performed by SpekPy, which can also take into account the operating exposure, given in the unit mA·s. In the following calculations, we input an exposure of 20 mA·s so that the fluence calculated by SpekPy corresponds to the fluence delivered by our setup during one second, also known as fluence rate $\dot{\phi}$, in cm$^{-2}$ s$^{-1}$, as per the ICRU definition [5]. Note that in some communities, this quantity is sometimes referred to as *flux*, or *flux density*, although ICRU prefers the use of the term *fluence rate*, to avoid confusion with other physical quantities [5].

This overall increase of fluence with operating voltage is also illustrated in Figure 5b, which depicts the evolution of the total fluence rate integrated over the whole spectrum with the operating voltage of the X-ray tube. As the graph indicates, the relation between voltage and fluence rate is almost linear, and can be roughly approximated in these conditions as voltage [kV] $\times 10^9$ [cm$^{-2}$ s$^{-1}$ kV$^{-1}$].

Using these simulated spectra, we can also determine their mean energy, which is a useful characteristic to compare different kinds of irradiation beams. This quantity can be calculated by considering the energy spectrum $\phi(E)$ as a continuous and unnormalized probability density function. Its mean energy $\overline{E}$ and mean energy-fluence $\overline{\psi}$ (considering

$\psi = \phi E$) are therefore the expected value of the corresponding distributions, which are given by Equations (1) and (2), and give the results shown in Table 3.

$$\bar{E} = \frac{\int E\phi(E)dE}{\int \phi(E)dE} \quad \text{for mean energy} \tag{1}$$

$$\bar{\psi} = \frac{\int E^2\phi(E)dE}{\int E\phi(E)dE} \quad \text{for mean energy-fluence} \tag{2}$$

**Table 3.** Mean energy and energy-fluence of the simulated X-ray beam at different voltages.

| X-ray Tube Voltage | 30 kV | 60 kV | 100 kV | 160 kV | 225 kV |
|---|---|---|---|---|---|
| Mean energy $\bar{E}$ [keV] | 12.2 | 17.2 | 25.1 | 37.4 | 49.5 |
| Mean energy-fluence $\bar{\psi}$ [keV] | 14.3 | 24.4 | 40.1 | 60.3 | 79.4 |

*3.3. Dose Sensitivity of Optical Fiber According to Energy or X-ray Tube Voltage*

Using the Monte-Carlo simulation process described in Section 2.2, we determined the dose/fluence response of the optical fiber and the ionization chamber by running simulations with two different types of source spectra. In the first case, we simulated a monoenergetic source, varying between 1 keV and 2000 keV with 50 energies spaced evenly in a logarithmic scale. In the second case, we considered an X-ray photon spectrum, taking as input the spectrum simulated by SpekPy (described in Section 3.2), with 40 voltages from 30 kV to 225 kV with a step of 5 kV. An important parameter to consider is that Geant4 normalizes any given source spectrum, and therefore the fluence determined by SpekPy has no influence on these results.

The energy response of the OF, in the form of the simulated dose over fluence calculated separately for core, cladding and coating, is shown in Figure 6a for monoenergetic photons and Figure 6b for X-ray tube spectra at different operating voltages. For these two figures, uncertainties at $\pm 2\sigma$ are lesser than 10%, which corresponds approximately to the thickness of the plot line with the chosen log scale of these graphs.

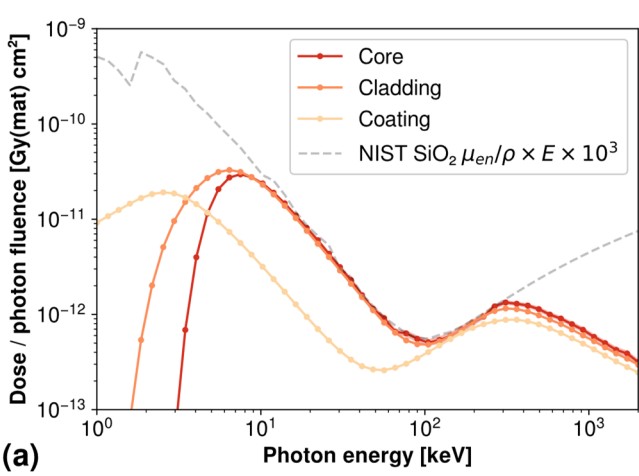
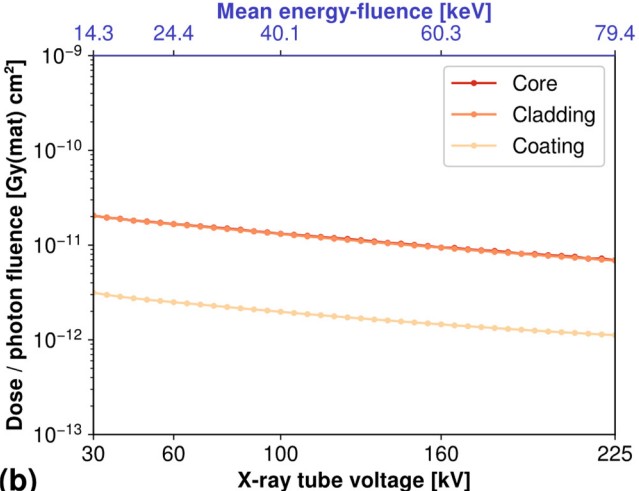

**Figure 6.** (**a**) Simulated dose/fluence of the OF for monoenergetic photons from 1 keV to 2 MeV. (**b**) Simulated dose/fluence of the OF for X-ray tube spectra at different operating voltages.

The monoenergetic response shown in Figure 6a features a high sensitivity to low energies around the 10 keV region, then a decreasing sensitivity with increasing energy, with the exception of a local maximum around the 300 keV region. The coating response appears overall lower than those of the core and cladding, except for very low energies,

which can be explained by the lower density and less interacting material simulated for the OF coating.

Core and cladding share a similar energy response, except for energies lower than 10 keV for which the cladding receives more dose than the core. This difference in very low energies between core, cladding and coating can be explained by the geometry of the OF (cf. Figure 3), and the fact that very-low energy particles are mostly stopped by the coating, then by the cladding, and are converted to deposited dose in these regions. On the other side of the graph, the decrease of deposited dose at higher energies, above 300 keV, is explained by the fact that secondary particles are generated with a high kinetic energy, enabling them to travel far beyond the limits of the OF without losing all their energy in the form of deposited dose.

These hypotheses are supported by the superimposition of the mass-energy absorption coefficient $\mu_{en}/\rho$ for silica (with $\rho$ the material density), calculated from the tabulated values given by NIST for single elements, in the unit $cm^2/g$ [46] and multiplied by the energy (in J) to give the theoretical kerma, i.e., the energy of released secondary particles per unit of mass [5]. This value is further multiplied by a factor of $10^3$ to consider the conversion from grams (from the unit used in the NIST data) to kilograms (as per the definition of the kerma unit in J/kg). As it appears in Figure 6a, the simulated dose/fluence ratio in the OF core and cladding matches perfectly the theoretical kerma between 10 keV and 300 keV, meaning that all energy released in the form of secondary particles ends up being deposited in these parts of the OF, whereas the differences in lower and higher energies are explained by the statements presented above.

The response of the OF as a function of X-ray tube voltage shown in Figure 6b is comparatively less structured, and can be thought as a weighted average of the monoenergetic values; as a result, core and cladding dose sensitivities are identical over the whole investigated range of X-ray tube voltages. The data show an overall decreasing sensitivity of the OF with increasing voltage, by a factor of approximately 3 between the dose/fluence ratio simulated at 30 kV and the one simulated at 225 kV.

### 3.4. Dose Sensitivity Spectrum of the Optical Fiber under X-rays

Combining the X-ray spectra simulated in Section 3.2 with the OF core monoenergetic dose response simulated in Section 3.3, we obtain the graph in Figure 7a showing the contribution of each photon energy to the total dose deposition in the OF. Uncertainties are estimated from both SpekPy and Geant4 simulations, and amount to approximately 10% over the whole spectrum.

Because the OF is more sensitive in the low-energy region around 10 keV (cf. Figure 6a), the contribution of the low-energy region of the X-ray spectrum is significantly enhanced compared to higher energies. In particular, the series of L characteristic lines appear to predominate the contribution to the total amount of dose deposited in the OF for all investigated X-ray voltages.

To further highlight this contribution of the low-energy part of the X-ray spectrum to dose deposition, Figure 7b shows the cumulative integral of the dose response spectrum in Figure 7a, normalized by the integral of this whole spectrum. Figure 7c shows the same data in a stacked bar plot format to further highlight the contribution of each energy bin to total dose deposition.

For all investigated voltages, at least 90% of the dose deposited in the OF is caused by photons with an energy lower or equal to 30 keV (respectively, 75% below 15 keV). Moreover, the contribution of the characteristic X-ray emission peaks is also very significant, with $L\alpha$ contributing from 26% to 30% to the total dose deposition, and $L\beta$ between 20% and 30%.

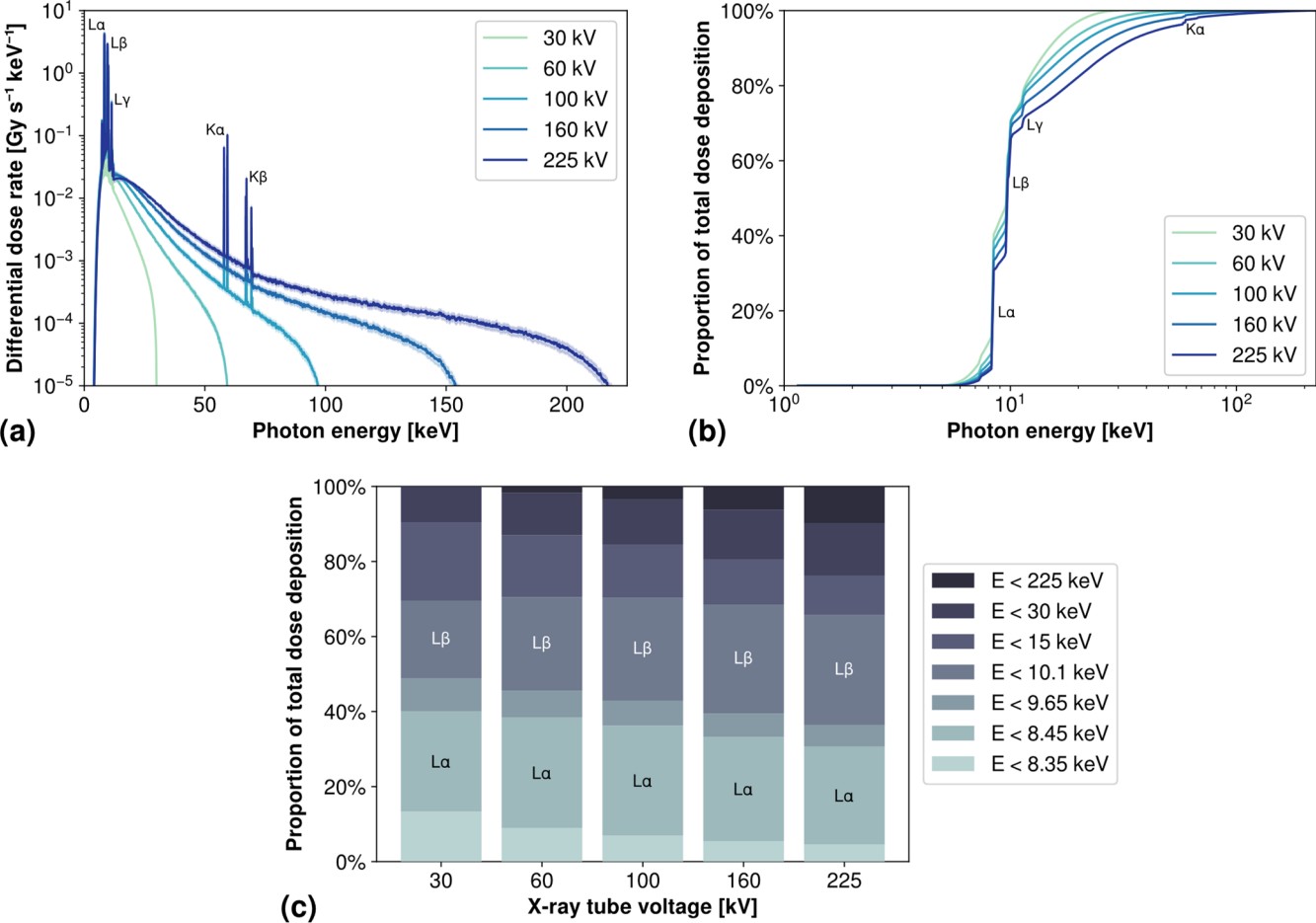

**Figure 7.** For five different source voltages between 30 kV and 225 kV: (**a**) Simulated dose deposition spectra of the X-ray beam inside the OF. The names of the main characteristic X-ray emission lines of tungsten are given next to the corresponding peaks. Filled areas indicate uncertainties at $\pm 2\sigma$. (**b**) Simulated cumulative dose deposition spectra of the X-ray beam, in an energy interval up to 30 keV. The name of the tungsten characteristic X-ray lines contributing the most to dose deposition are shown next to the corresponding sharp increases. (**c**) Stacked bar plot of simulated cumulative dose response showing the contribution of each energy interval to the total dose. Contributions due solely to characteristic tungsten X-ray lines L$\alpha$ and L$\beta$ are indicated inside the corresponding bins.

*3.5. Dosimetry Measurements Using Optical Fiber*

Using the experimental setup described in Section 2.1, we acquired the intensity transmitted through OF samples using different types of measurements, and calculated the RIA by applying the following formula:

$$\text{RIA} = -10 \log_{10} \left( \frac{I - I_{\text{dark}}}{I_0 - I_{\text{dark}}} \right)$$

with $I$ the measured intensity at each instant, $I_0$ the intensity at irradiation start, and $I_{\text{dark}}$ the intensity with the light source switched off.

Figure 8a shows the spectral RIA, in the infrared range, of the OF acquired after irradiating up to 3000 s with an operating voltage of 100 kV. It features a clearly defined band around the 1550 nm region, which is known to be the signature of the P$_1$ defects of P-doped OFs, that present an optical absorption band peaking at 0.79 eV [20] and offer a great interest for dosimetry, as introduced in Section 1. As shown in the inset, RIA varies very little between 1500 nm and 1600 nm, and the deviation between values measured hereafter at 1532 nm and 1550 nm is lesser than 0.5%.

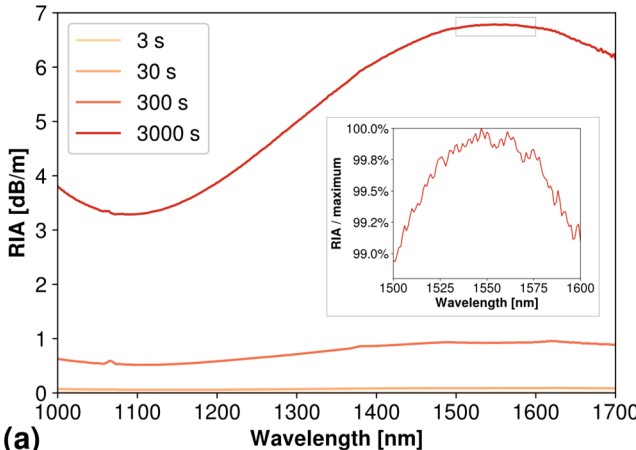
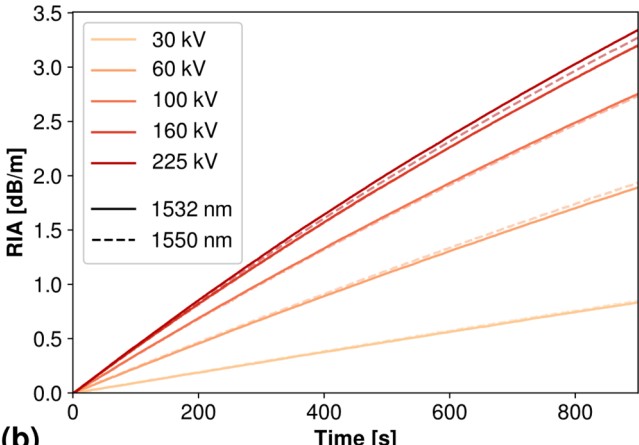

**Figure 8.** (**a**) Spectral RIA of the optical fiber irradiated during 3, 30, 300 and 3000 s with a tube voltage of 100 kV. Inset magnifies the framed area at 3000 s between 1500 nm and 1600 nm, normalized by maximum. (**b**) Evolution of optical fiber RIA at 1532 nm and 1550 nm according to irradiation time, at different X-ray tube voltages.

Figure 8b shows the evolution of the RIA measured at 1532 nm and 1550 nm according to irradiation time, for five operating voltages between 30 kV and 225 kV. Evolution of RIA with time is close to linear, and shows a very similar trend between different operating voltages. By performing a linear fit on the first 100 s of each of these RIA measurements, we obtain the rate of OF RIA increase for each investigated operating voltage, summarized in Table 4.

**Table 4.** Experimental RIA rates of OF samples at 1532 nm and 1550 nm over the first 100 s of irradiation, for investigated X-ray tube voltages. Uncertainty is estimated to be less than ±1%.

| X-ray Tube Voltage | 30 kV | 60 kV | 100 kV | 160 kV | 225 kV |
|---|---|---|---|---|---|
| RIA rate at 1532 nm [dB m$^{-1}$ s$^{-1}$] | 0.93 | 2.32 | 3.52 | 4.20 | 4.41 |
| RIA rate at 1550 nm [dB m$^{-1}$ s$^{-1}$] | 0.97 | 2.35 | 3.49 | 4.20 | 4.32 |

For both investigated wavelengths, because of the high linearity of the measurements and the high number of measurement points, the uncertainty in the values given in Table 4 is estimated to be less than 1%.

### 3.6. Dosimetry Measurements Using Ionization Chamber

To check the response of a conventional dosimetry system with different X-ray energies, we placed the ionization chamber described in Section 2.1 at four locations placed 5 cm around the center of the irradiated OF for each voltage (see Figure 4b).

The results of the dose rate measurements using this method are shown in solid lines on Figure 9, in the unit Gy(H$_2$O)/s which is the one displayed by the dosimetry device. These data show a clear increase of the dose rate perceived by the ionization chamber with increasing tube voltage, consistent with the simulated increase of beam fluence rate reported in Figure 5a. However, the rate of increase at higher energies is visibly reduced for the ionization chamber, whereas beam fluence appears to follow a more linear trend through all investigated voltages.

Measurements taken through the four dosimetry locations are in good accordance, and show a good beam homogeneity at the locations corresponding to the irradiated OFs. However, point B stands out as its data significantly deviates from the other points at higher operating voltages, whereas point A, which stands symmetrical to point B in the x-axis, appears relatively unaffected. This deviation can first be explained by a slightly incorrect estimation of the experimental location of the beam center, along with the practical

uncertainty of ±1 cm, considering the positioning inaccuracy of the ionization chamber given its large dimensions.

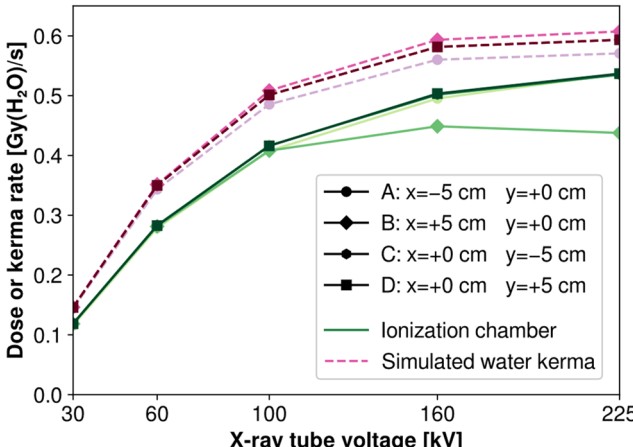

**Figure 9.** Dose rate measured by the ionization chamber (solid line) and simulated water kerma rate (dashed line) at five tube voltages and four points in a radius of 5 cm around locations corresponding to the center of the irradiated OFs.

Another parameter to take into account is the global decrease of beam homogeneity in the x-axis at higher voltages, as the beam takes a more elliptical shape (cf. Section 3.1). By calculating the theoretical water kerma rate from the simulated spectra at each voltage and location, along with the NIST $\mu_{en}/\rho$ data for water [46], we can reproduce to an extent the deviation observed on the x-axis, as shown in dashed lines on Figure 9, for a center of the dosimetry locations perfectly located on the beam maximum.

Comparison between experimental and simulated dosimetry values shows a systematic over-estimation of the dose rate by simulation compared to the experiment. Because the dose rate in water given by the ionization chamber is supposed to be at electronic equilibrium and therefore equal to kerma, both these quantities should be comparable.

This observed deviation can be explained by the difference between the fluence simulated by SpekPy and our actual experimental setup, and we can therefore estimate a factor $k_{simul} = 0.83$ (±10%) cm$^{-2}$/cm$^{-2}$ to take this difference into account.

*3.7. Comparison between Simulation, Optical Fiber and Ionization Chamber*

Figure 10 summarizes all measurements and simulations performed in this work by displaying side-by-side the dose rate estimated by three different means.

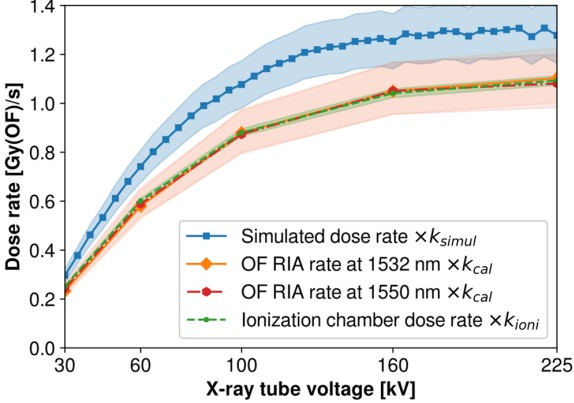

**Figure 10.** Comparison between dose rate determined by three different means: simulation, scaling of RIA rate and scaling of ionization chamber measurements; for X-ray tube voltages between 30 kV and 225 kV. Filled areas show uncertainties at ±2σ.

The simulated dose/fluence in the OF core obtained in Section 3.1 was multiplied by the fluence rate of the X-ray beam (as shown in Figure 5a) to obtain the simulated dose rate in the OF core for each operating voltage. This simulated dose rate was further multiplied by the factor $k_{simul}$, determined in Section 3.6, to consider the difference between simulated and experimental fluences in our setup.

The OF RIA rates determined in Section 3.5 were multiplied by the calibration factor $k_{cal} = 4.0 \times 10^{-3}$ ($\pm 10\%$) dB km$^{-1}$ Gy$^{-1}$, determined under $\gamma$ rays in a previous experiment involving the same type of OF [15], to obtain an estimated dose rate as measured by the OF at the different experimented voltages.

Finally, to compare the dose rate in water measured locally by the ionization chamber in Section 3.6 with these other results specific to the dose deposited in the OF, we used a fit algorithm to determine a simple relation between displayed ionization chamber dose rate and dose rate estimated by OF RIA. These calculations show that a very simple linear relation exists between these two measurements, with a constant scaling factor defined here as $k_{ioni} = 2.14$ ($\pm 10\%$) Gy(OF)/Gy($H_2O$) for all investigated voltages. The very good accordance between both these measurements is further highlighted by the superimposition of these two trends in Figure 10.

We observe that although all experimental results show a very good agreement when scaled as described above, simulation results appear to be systematically overestimated, by a factor of approximately 1.2, slightly varying with voltage. As this discrepancy cannot be explained by the difference between simulated and experimental fluences which is already taken into account by the factor $k_{simul}$, other factors can be considered to explain these deviations, such as uncertainties on OF positioning, OF sample lengths, OF dimensions and composition, and/or specifications of the X-ray tube.

## 4. Conclusions

By combining simulation tools and experimental work, we investigated the effects of varying the operating voltage of an X-ray tube on the dose deposition in a radiosensitive P-doped OF.

A first important effect of modifying the tube voltage is a shift of the X-ray beam center along the x-axis, with a span of about 5.5 cm over the investigated range of voltages, between 30 kV and 225 kV. This parameter has to be taken into account to properly position irradiated samples around the actual center of the beam.

Simulated X-ray spectra using SpekPy show the relative weights of bremsstrahlung and characteristic emission, and highlight the increase of X-ray photon fluence with increasing operating voltage over the whole spectrum, even at lower energies.

Monte-Carlo simulations of the monoenergetic dose/fluence response spectrum of the OF using Geant4 show that both core and cladding are most sensitive to photons around 10 keV, and reach electronic equilibrium between 10 keV and 300 keV. Photons of energy lower than 10 keV are unable to penetrate the fiber coating, and photons of energy higher than 300 keV release secondary electrons of too high a velocity to have their energy fully converted to deposited dose. Polyenergetic dose/fluence response shows that due to this higher contribution of the low-energy part of the spectrum to dose deposition, X-ray spectra of lower voltages tend to deposit more dose inside the OF; although this phenomenon is largely counterbalanced by the observed increase of fluence with operating voltage described above.

The important contribution of the low-energy part of the spectrum to OF dose deposition is further highlighted by combining both simulation results, which show that 90% of the dose is deposited by photons under 30 keV. These results also emphasize the very significant influence of tungsten characteristic X-ray emission peaks L$\alpha$ and L$\beta$, which, combined, amount to between 47% and 57% of the total deposited dose through all investigated voltages.

These simulation results were compared with experimental work to assess the fidelity and reliability of the proposed simulation toolchain. Two kinds of measurements were

acquired: RIA of P-doped OF and water dose rate using an ionization chamber. All these experimental measurements scale very well, and we determined a scaling factor of 2.14 ($\pm$10%) Gy(OF)/Gy($H_2O$), practically constant for all investigated voltages, to convert the ionization chamber dose rate to the actual dose rate perceived by the OF, which demonstrates the ability of phosphosilicate OFs to be used accurately as X-ray dosimeters through different tube voltages. Overall, simulation and experimental results estimate a factor of approximately 4.5 between dose rates at 30 kV and 225 kV, for the same operating current. This shows that in the case of OF irradiation, the range of available dose rates for a given X-ray irradiator can be extended by proper adjustment of the operating voltage. Moreover, dosimetry performed using conventional tools, such as the ionization chamber studied here, appears to deliver consistent results over the whole investigated range of voltages, and therefore the use of a single scaling factor for all voltages appears to be appropriate.

Dose rate estimated by simulation displays the same increasing trend with operating voltage as experimental results, but is still slightly over-estimated by a factor of approximately 1.2. This shows the ability and limitations of the proposed simulation toolchain to estimate, to some extent, the absolute value of experimental results.

Investigations on the causes of inaccuracy in the estimation of the experimental dose rate could be a subject for further research, so that a complete dosimetry model could be produced for a given X-ray irradiator, enabling an efficient planning and optimization of sample irradiation.

Moreover, the overwhelming influence of the low-energy photons on OF dose deposition suggests that beam filtering—using for example aluminum which is known to attenuate the lower part of the spectrum—could produce very significant changes in the dosimetry. This could be used both to reach lower dose rates and to favor certain physical processes, such as Compton scattering, because of the higher mean energy of the X photons in this case. The effects of such filtering at different X-ray voltages could be the topic of further research, along with an exploration of higher voltages and photon energies.

Overall, these results broaden the knowledge of the sensitivity of P-doped OF dosimeters under X-ray beams, and outline their practical advantages and limits. The present work has shown that such devices can be used with a reliability comparable to conventional ionization chambers at X-ray tube voltages from 30 kV to 225 kV. On the other hand, their high sensitivity to the low-energy part of the spectrum may deviate their response from standard dosimetry units, such as dose in water, in case of higher energy beams. This behavior at higher energies could be further assessed and investigated, although the good accordance demonstrated here between simulation results and $^{60}$Co $\gamma$ ray calibration performed in [35] hints toward the reliability of this dosimetry technique at photon energies in the MeV range, at least when the low-energy part of the spectrum is negligible. Finally, the use of optical attenuation as a means of measurement involves an appropriate length of OF, especially when a high sensitivity is required, which can circumvent the dimensional advantages of using an OF to perform localized dosimetry; in which case, more localized techniques such as radioluminescence-based fiber dosimeters could provide an advantage.

**Author Contributions:** Conceptualization, S.G., A.M. (Adriana Morana) and A.M. (Arnaud Meyer); methodology, S.G., A.M. (Adriana Morana), D.L. and A.M. (Arnaud Meyer); software, A.M. (Arnaud Meyer); formal analysis, A.M. (Arnaud Meyer); writing—original draft preparation, A.M. (Arnaud Meyer); writing—review and editing, A.M. (Arnaud Meyer), S.G., A.M. (Adriana Morana) and D.L.; supervision, S.G., D.L., P.P. and A.B. All authors have read and agreed to the published version of the manuscript.

**Funding:** This research received no external funding.

**Institutional Review Board Statement:** Not applicable.

**Informed Consent Statement:** Not applicable.

**Data Availability Statement:** The data supporting the findings of this study are available from the corresponding authors upon reasonable request.

**Conflicts of Interest:** The authors declare no conflict of interest.

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
