# Peer review of "Simulation and Optimization of Optical Fiber Irradiation with X-rays at Different Energies"

_radiation, doi:10.3390/radiation3010006_

Round 1

Reviewer 1 Report

The aim of this work regards the Simulation and Optimization of Optical Fiber Irradiation with X-Rays at Different Energies.

Optical fibers are nowadays of promising interest for dosimetry.

This work is potentially interesting even if there are some details needing some clarifications:

1. page 1-2 - lines 41-46 “A typical X-ray ...an electron beam” This sentence is trivial, I suggest to remove it.

2. page 4 – Figure 2: This Figure is not clear, in particular the blu fiber is not a spiral as described at page 4, line 139.

3. page 7-8 lines 283-285 “Note that in this article, we preferred the term fluence rate, ...instead of flux ...expressed in s-1”. I strongly suggest to remove or re-write this very confusing sentence, in your paper, "fluence rate" is the only correct term you can use.

4. page 9, line 349 “...we obtain the graph in Figure a, showing...”. Please add here the number of this figure.

5. page 9, line 360 “...in Figure a, normalized by the integral of this whole spectrum. Figure c shows...”. Please add the number of Figure a and Figure c.

6. page 11, line 411 “The results of the dose rate measurements using this method are shown in Figure 9a...”. As described at page 4 lines 160-162 the PTW 23344 is a ionization chamber optimized for use with X-ray tubes voltages from 15 kV to 70 kV. In your work the voltage range is 30-225 kV, so mostly outside the optimal range of this chamber: How did you consider this? Did you use any correction factor to take it into account? Please specify.

Reviewer 2 Report

Paper review “Simulation and Optimization of Optical Fiber Irradiation with X-Rays at Different Energies”

The authors of the paper present the influence of the voltage of an X-ray tube on the total ionizing dose deposited in radiation-sensitive optical fibers. In addition to experimental data, they present a toolchain of different simulation packages and show that it is suited to model dose deposition in optical fibers.

Please find my comments, which should be addressed, listed as bullet points below.

·       The Introduction has only very few references and the importance of accurate dosimetry for various applications, especially in a medical context, should be highlighted there

·       The term “X photons” is not commonly used, “X-rays” would be better and should be used consistently throughout the whole text

·       Line 83: the “P1 defect” should be explained since it is not mentioned before

·       Line 93: the terms “Total Ionizing Dose (TID)” and “Displacement Damage Dose (DDD)” should be explained

·       In general, the concept of dose should be explained in a bit more detailed since not all readers will be familiar with the concept

·       Section 2: what is the size of the X-ray tube focal spot?

·       Figure 1: it could be helpful to also include the cathode in the schematic and also indicate at which positions in the total fluence rate is calculated

·       Lines 153 and 154: where is the beam maximum and where is Section 0?

·       Why can the ionization chamber also be used for higher voltages if it is not optimized there? What does this mean in terms of interpretation of the results?

·       Line 178: see above, where is Section 0? Should be corrected in all parts of the manuscript

·       Line 197: where do the source dimensions come from? Are those the values of the actual tube focal spot?

·       Line 266: what is the physical reason for the observed intensity increase?

·       Table 2: why are two lines mentioned but only one mean energy?

·       Figure 5b): why are the uncertainties higher at higher voltages?

·       Line 349, 359, 360: number of the Figure is missing

·       How trustworthy are the results even though the response of the OF and the ionization chamber is only characterized for low X-ray energies?

·       Line 438: can this factor be motivated?

·       The explanation for the overestimation by simulations is not very scientific and a more detailed error discussion should be provided, maybe underlined with additional simulation or experimental results or numerical estimates

·       Are all effects inside the dosimetry devices accurately modelled in the simulations in order to allow a quantitative comparison to experimental data? It would be helpful to add more details about the dose calculations in the Geant4 simulations.

·       Formatting of the references should be made consistent/numbers need to be checked

Reviewer 3 Report

The authors have studied the energy dependence of incident photons on an optical fibre based compact dosimeter. While the work itself is performed in a good scientific manner, the authors do not provide convincing evidence on the impact of their work as detailed below. Besides this overall critique, here comes my list of critical points that need to be address for any re-submission.

line 94: they only cite a paper on DDD, but should spend a more detailed sentence on what DDD actually is.

line 197: it is not explained why these source dimensions have been selected.

lines 201 ff: a detailed explanation on how the dose was calculated with G4 needs to be included. For instance, was this calculated voxel-wise or just for the entire volume and how was this done?

line 250 and elsewhere: it is often referred to a "section 0", but there is no such section.

line 250/257: it is not clear at this point how the simulated object really looks like: is it just the fibre or is this fibre embedded in some larger object?

line 289: why is the dependence just "almost linear" and not linear as expected?

fig. 6b: I would add second x-axis showing the mean energy for better comparison with 6a.

lines 353ff: this is the main critical point: if the fibers are most sensitive around 10 keV only, how can this dosimeter technique be used for more energetic X-ray beams? In radiation therapy much higher photon energies occur. The authors fail to give a good account on that matter.

fig. 9a+9b: it would be better to show all curves in one and the same plot for better direct quantitative comparison.

In the conclusion: besides my critical point raised above already, there is yet another issue that needs to be addressed: what are the practical limitations as the optical fibers need also input/output fibers, that is, how can this be incorporated inside a test object (if it is not water, but a solid phantom, for instance)?

Overall, the current manuscript is valid in its details and the scientific method used, but the conclusion is not convincing overall, hence I strongly recommend to focus more on there practical aspects and possible impact: where can this method be used, where would it be impractical?

Round 2

Reviewer 1 Report

Authors mostly answered questions and improvements suggested.

Reviewer 2 Report

The authors have provided detailed answers and made important improvements to the manuscript. From my point of view, it can be accepted in the present form.